# Hair Dye and Relaxer Use among Cisgender Women in Embu and Nakuru Counties, Kenya: Associations with Perceived Risk of Breast Cancer and Other Health Effects

**DOI:** 10.3390/ijerph21070846

**Published:** 2024-06-28

**Authors:** Adana A. M. Llanos, Adiba Ashrafi, Teresa Olisa, Amber Rockson, Alexis Schaefer, Jasmine A. McDonald, Mary Beth Terry, Dede K. Teteh-Brooks, Dustin T. Duncan, Beatrice Irungu, Cecilia Kimani, Esther Matu

**Affiliations:** 1Department of Epidemiology, Mailman School of Public Health, Columbia University Irving Medical Center, New York, NY 10032, USA; ai2337@cumc.columbia.edu (A.A.); ar4571@cumc.columbia.edu (A.R.); as7320@cumc.columbia.edu (A.S.); jam2319@cumc.columbia.edu (J.A.M.); dd3018@cumc.columbia.edu (D.T.D.); 2Herbert Irving Comprehensive Cancer Center, Columbia University Irving Medical Center, New York, NY 10032, USA; 3African-Caribbean Cancer Consortium (AC3), Philadelphia, PA 19111, USA; 4Centre for Traditional Medicine and Drug Research, Kenya Medical and Research Institute, Nairobi, Kenya; lysaterry3@gmail.com (T.O.); birungu18@gmail.com (B.I.); ceciliakmn@yahoo.com (C.K.); 5Department of Health Sciences, Crean College of Health and Behavioral Sciences, Chapman University, Orange, CA 92866, USA; teteh@chapman.edu

**Keywords:** hair products, hair dyes, chemical relaxers, perceived breast cancer risk, perceived health risks, cisgender women, Kenya

## Abstract

Despite widespread use of hair products globally, little is known about the prevalence and patterns of use in populations outside the United States. As some hair products contain endocrine-disrupting chemicals (EDCs) and EDCs have been linked to breast cancer, which is increasing globally, in this study, we addressed key knowledge gaps about hair product use and practices, and perceptions of use among women in two counties in Kenya. Using community-engaged approaches in Embu and Nakuru, Kenya, we recruited women aged 15–50 years to complete a questionnaire that ascertained hair product use in the last 7–14 days, ever using hair dyes and chemical relaxers, and participants’ perceptions or harm around hair product use. In multivariable-adjusted regression models, we evaluated associations between participants’ sociodemographic characteristics and perceptions of hair product use in relation to if they have ever used hair dyes and relaxers. In our sample of 746 women (mean age, 30.4 ± 8.1 years), approximately one-third of participants reported ever using permanent and/or semi-permanent hair dyes, with approximately one-fifth reporting current use. Almost 60% reported ever using chemical relaxers, with a little over one-third reporting current use. Increasing age and having an occupation in the sales and service industry were statistically significant predictors of hair dye use (OR 1.04, 95% CI: 1.02–1.06 and OR 2.05, 95% CI: 1.38–3.03, respectively) and relaxer use (OR 1.03, 95% CI: 1.01–1.06 and OR 1.93, 95% CI: 1.30–2.87). On average, participants reported moderate-to-high levels of concern about exposures and general health effects from using hair products, and relatively high levels of perceived risk of breast cancer related to hair product use. However, in contrast to our hypotheses, we observed mixed evidence regarding whether higher levels of perceived risk were associated with lower odds of ever using hair dyes and relaxers. These findings add new knowledge to the extant literature on hair product use among women in Kenya, where breast cancer incidence rates are increasing. Improving the understanding of patterns of use of specific products and their chemical ingredients—which may be hormone disruptors or carcinogens—and exploring the role of environmental health literacy are critical for developing interventions to reduce potentially harmful exposures found in these products.

## 1. Introduction

Breast cancer is the most frequently diagnosed cancer and the second leading cause of cancer-related death among women in Kenya [1]. With over 6500 new cases and 2500 related deaths each year, breast cancer proves to be a substantial burden on this population [2]. Based on current trends, these rates will continue to rise with an estimated 12,300 new cases and 6000 deaths annually by the year 2035 [2]. The rising rates of breast cancer have been associated with an increased prevalence of risk factors linked with the economic transition including obesity, physical inactivity, reproductive behaviors, and smoking [3]. The increased incidence has also been attributed to greater awareness and detection of breast cancer as well as the growth of an aging population [3,4,5].

Emerging evidence in countries including the United States (US) and Ghana show that use of personal care products (PCP), particularly hair products, may be associated with an increased risk of breast cancer and potentially with more aggressive breast tumor phenotypes among Black/African ancestry (B/AA) women [6,7,8,9]. These associations are supported by evidence connecting endocrine-disrupting chemicals (EDCs) and other toxic chemicals, including carcinogens and mutagens, with adverse health outcomes and their presence in mainstream hair care products [10,11,12]. In an analysis of the Sister Study, a large prospective cohort study in the US, investigators evaluated the relationship between hair dye and breast cancer using self-reported questionnaires and annual health updates with medical records as confirmation of diagnosis in 46,709 women [7]. The participants that used permanent dyes or relaxers/straightener products in the 12 months prior to study enrollment were at a higher breast cancer risk [7]. This association was especially apparent among B/AA participants, who were found to have a 45% higher breast cancer risk than those who did not use dye or straighteners [7]. A case-control analysis in the Women’s Circle of Health Study in a sample of B/AA and white women in New York and New Jersey (N = 4285) reported similar associations between hair dye and increased breast cancer risk [9], and subsequent findings showed associations with longer duration of hair dyes and relaxers with breast tumor clinicopathologic features including tumor grade and size, receptor status, and lymph node status, which were not previously explored [8]. While no studies have examined the associations of PCP and hair product use with breast cancer among women in Kenya, findings from a relatively small study in Nakuru, Kenya (N = 242) showed a prevalence rate of 59% for chemical relaxer use, with long-term use (4 years) reported among 41% of the sample studied [13].

The use of PCPs is prevalent among women of all races and ethnicities [10,11,12,13,14,15,16], though most studies have been conducted in the US. In the Taking Stock Study, which included a racially and ethnically diverse sample of women in California (N = 357), investigators observed an average of eight PCPs are used each day among participants [14]. In our cross-sectional analysis of PCP use among adults at an academic institution, we found on average, women used 19 PCPs per day [15]. From the Taking Stock Study and our own analysis, we note that while most cisgender women use PCPs, the types of products and frequency of use varies. For example, B/AA women tend to use more hair products [14,15]. This contributes to an unequal burden of exposure to environmental chemicals, including EDCs among women of color [11], which may be linked to inequities in adverse health outcomes [11,16], including risk and outcomes of breast cancer.

Building on our prior work [15,17], the current study sought to evaluate the prevalence and patterns of PCP use and perceptions and attitudes around use in an understudied international, low-resource setting, Kenya—where the current National Cancer Control Strategy includes reducing exposure to environmental and occupational risk factors [2]. The primary objectives of this study were to examine the use of PCPs, with a focus on hair products among cisgender women in two counties in Kenya (Embu and Nakuru), and to assess attitudes and perceptions about the potential health risks (including perceptions of breast cancer risk) associated with using these products. This manuscript—the first of a series from our ongoing work—serves as an introduction to a newly established binational research collaboration focused on understanding the prevalence and patterns of PCP and hair product use (and perceptions about the health implications of exposure to chemicals in these products) among women in Kenya. Findings from this study will also inform the development of interventions to reduce chemical exposures among vulnerable groups (e.g., adolescent girls, pregnant people, sexual and gender minority populations, breast cancer survivors, and those who may be at increased risk of breast cancer and other hormone-related cancers).

## 2. Materials and Methods

### 2.1. Participants and Setting

This study collected self-reported sociodemographic information knowledge, attitudes, and perceptions about PCP use, and use of specific PCPs and hair products in a sample of women aged 15–50 years in Embu and Nakuru Counties in Kenya between May and July 2023. We sampled cisgender women from Nakuru County (estimated population of 570,674 people and ranks at #3 regarding gross domestic product [GDP]) and Embu County (estimated population of 608,599 and ranks at #24 regarding GDP). These counites are separated by a travel distance of 156 km. Prospective study participants were identified using a combination of convenience sampling of randomly selected beauty shops/salons and purposive sampling of households in select sub-counties of Nakuru County (Nakuru West, Nakuru East, Bahati, and Naivasha) and Embu County (Embu North, Embu South, Embu East, and Mbeere). Nakuru County is more urbanized (46%), has a population with widely diverse ethnicities (representing a heterogeneous population), and has a breast cancer prevalence rate of 6.9 per 100,000 [18], while Embu is less urbanized (16%), has a more homogeneous local population (representing primarily Embu, Kamba, and Mbeere ethnicities), and has a breast cancer prevalence rate of 31.9 per 100,000 [18]. Given these population differences, we hypothesized potential heterogeneity in PCP and hair product use and perceptions of use.

Field research team members—which included local public health officers and community health volunteers—approached prospective participants present in the shops/salons or selected households and verbally provided information about the study. Individuals who were eligible and interested in participation provided written consent prior to completing the interviewer-administered study questionnaire, which was available in English and Kiswahili and took approximately 20–30 min to complete. Upon completion of the questionnaire, participants were given a reimbursement of KES 600 (equivalent to approximately USD 5) in appreciation of their time. This study was reviewed and approved by the Kenya National Commission for Science, Technology, and Innovation (NACOSTI), the Scientific and Ethics Review Unit of the Kenya Medical Research Institute (KEMRI), and the Institutional Review Board of Columbia University.

### 2.2. Data Collection

For this study, we used a modified version of the Personal Care Product (PCP) Use Questionnaire [15]. Some response options were modified to be culturally relevant to the Kenyan context (e.g., categories of the sociodemographic questions were changed for greater relevance). The product use questions were modified to assess product use in the last 7–14 days with a slightly greater focus on hair products. Lastly, we included the 6-item Black Identity, Hair Product Use, and Breast Cancer Scale (BHBS) to assess perceived breast cancer risk related to hair product use [19]. In this modified version of the PCP use questionnaire, there was a total of 64 questions in six sections: (A) getting to know you (i.e., sociodemographic information and health history), (B) choosing/buying hair care products, (C) perceptions and attitudes about personal care and hair products, (D) hair product use, (E) use of other personal care products, and (F) additional questions on hair product use, including the age at which hair dye and relaxer use began (≤12, 13–19, ≥20 years).

Sociodemographic characteristics queried included age (years), religion (Christian, Muslim, other), marital status (currently married, formerly married [divorced, separated, widowed], never married), education level (less than high school certificate, high school certificate, some college but no degree, bachelor’s degree and above), current occupation field (business and administration, education, government, law enforcement, farming and agriculture, sales and service, science, technology, engineering, student, unemployed, or casual work), current monthly income (KES <10,000, KES 10,000–50,000, KES >50,000), household size (1–2, 3–4, ≥5), and number of children under age 15 residing in household (0, 1, 2, ≥3). Health history characteristics queried included age at menarche (years), menopausal status (premenopausal, postmenopausal), personal history of breast cancer (yes, no), and current pregnancy status (yes, no).

Perceptions about health effects related to hair product use were assessed by asking participants to indicate their level of agreement (5-point Likert scale: strongly agree, agree, neither agree nor disagree, disagree, or strongly disagree) with six statements: (1) “The personal hair care products I use affect my health”, (2) “Organic, natural, non-toxic, or eco-friendly personal hair care products have fewer toxic chemicals than regular products”, (3) “Consumers should be concerned about the health effects of personal hair care products”, (4) “There is no reason to worry about the health effects from chemicals that might be in personal hair care products”, (5) “Overall, the benefits of using personal hair care products outweigh any risks from exposure to toxic chemicals that might be in these products”, and (6) “Government agencies in Kenya do a good job of regulating personal care products to ensure they are safe for consumers”. Based on the distribution of responses in the overall sample, item responses were dichotomized to reflect greater perceptions of harm/concern versus lower perceptions of harm/concern towards hair product use (e.g., “strongly agree” versus all other response options for statements 1–3, and “strongly disagree” versus all other response options for statements 4–6).

Perceptions about breast cancer risk associated with hair product use was examined using the BHBS [19], which queried participants’ level of agreement (5-point Likert scale: strongly agree, agree, neither agree nor disagree, disagree, or strongly disagree) with the following six statements: (1) “I am concerned that the labels of hair care products do NOT list all the ingredients”, (2) “Because I am concerned about breast cancer, I plan to go natural (style my hair without chemicals)”, (3) “Because I am concerned about breast cancer, I intend to watch the ingredients of the hair care products I use”, (4) “All women should worry about the ingredients in hair products”, (5) “Because I am concerned about breast cancer, I plan to adjust how I use hair care products”, and (6) “I want to learn more about the risk hair products can cause to my health”. Item responses were again dichotomized to reflect greater perceptions of harm/concern versus lower perceptions of harm/concern towards hair product use (e.g., “strongly agree” versus all other response options for all 6 statements).

A total of 746 study participants completed the study questionnaire, which represents 97.1% of our target enrollment (372 [96.9% of target] in Embu and 374 [97.4% of target] in Nakuru. Data collected on these individuals are included in the analysis presented herein.

### 2.3. Outcomes

The primary outcomes were ever using hair dyes and ever using relaxers. Ever using hair dyes was defined as the use of permanent and/or semi-permanent hair dye(s) more than twice per year for ≥1 year (yes, no). This was derived from participants’ responses to two questions: (1) “Have you ever regularly dyed your hair with permanent hair dye (including highlights) for at least one year? By regularly, we mean that you used permanent hair dye (including highlights) more than two times per year”, and (2) “Have you ever regularly dyed your hair with semi-permanent hair dye for at least one year? By regularly, we mean that you used semi-permanent hair dye more than two times per year”. Ever using relaxers was defined as the use of chemical relaxer(s) or straightening product(s) more than two times per year for ≥1 year (yes, no).

Secondary outcomes were current hair dye use and current relaxer use. Current hair dye use was defined as use of permanent and/or semi-permanent hair dye(s) in the past year (yes, no). This variable was derived from participants’ responses to two questions: (1) “Do you currently dye your hair or have you in the past 12 months dyed your hair using permanent hair dye?” and (2) “Do you currently dye your hair or have you in the past 12 months dyed your hair using semi-permanent hair dye?”; current relaxer use was defined as the use of chemical relaxer(s) or straightening product(s) in the past year (yes, no).

Additional details on hair product use were ascertained, including the age at which hair dye and relaxer use began (≤12, 13–19, ≥20 years), shade of hair dye used most frequently (light, medium, dark), type of relaxer typically used (no-lye, lye-containing, both, I do not know/I cannot remember), typical application of hair dye and relaxer products (at-home kit, salon, both), and number of different brands of hair dye and relaxer products used (1, ≥2).

### 2.4. Statistical Analysis

Sociodemographic and health history characteristics of study participants were compared across counties using chi-squared tests and analysis of variance (ANOVA) for categorical and continuous variables, respectively. Logistic regression models were run with robust error variances to estimate odds ratios (ORs) and their corresponding 95% confidence intervals (CIs) to explore the associations between participants’ perceptions about health effects and perceived breast cancer risk related to hair product use with ever using hair dyes, ever using relaxers, current use of hair dyes, and current use of relaxers. Models were initially run in the overall sample (N = 746), and then separately for Embu County (N = 372) and Nakuru County (N = 374). First, models explored associations between relevant sociodemographic and health history characteristics (age, education level, current occupation field, number of children under age 15 residing in the household, and age at menarche) with hair dye use and relaxer use. Then, scale items for perceptions about health effects and perceived breast cancer risk related to hair product use were added to the final models.

The Benjamini–Hochberg (BH) approach was employed to correct for multiple comparisons, using a false discovery rate (FDR) of 0.05. Following correction, descriptive analyses with *p*-values ≤ 0.008 and logistic regression analyses with *p*-values ≤ 0.002 were deemed statistically significant. SPSS (version 26.0) and STATA (version 16.1) were used for descriptive statistics and regression analyses, respectively.

## 3. Results

### 3.1. Participant Characteristics

Characteristics of the study participants are shown overall and by county in Table 1. Overall, the mean age was 30.4 ± 8.1 years and almost all participants reported their religion as Christian (99.1%). Most were either currently married (46.6%) or never married (46.1%), not pregnant (92.2%), premenopausal (87.7%), and had no personal history of breast cancer (97.3%). In terms of education, 19.0% had less than a high school education, 37.9% had a high school education, 33.9% had some college education but no degree, and 9.0% had a bachelor’s degree or above. Participants’ occupations differed significantly by county (*p* < 0.001). Significantly larger proportions of individuals from Embu reported employment in business and administration (26.1% vs. 15.5%) and being unemployed or working casual jobs (9.4% vs. 3.7%) relative to individuals from Nakuru, while those from Nakuru more frequently reported employment in sales and service occupations relative to Embu (55.9% vs. 41.7%), the majority of whom were cosmetologists (50.8% vs. 30.1%). Individuals from Nakuru also reported higher monthly incomes (*p* = 0.003) and having more children under the age of 15 years in the household (*p* = 0.002) than those from Embu.

### 3.2. Prevalence of Hair Dye and Relaxer Use

Approximately one-third of participants (31.1% and 37%) had used permanent and semi-permanent hair dyes at some point in their lives, respectively, and 17.6% and 21.3% reported currently using permanent and semi-permanent hair dyes, respectively (Table 2). Most (70.3%) reported initiating hair dye use at ≥20 years (Table 2). Few differences were noted in hair product use by county, including the shades typically used, typical application type, and current use. More than half of participants had used relaxers at some point in their lives (59.4%) and a little over one-third reported current use (35.7%), with 33.9% starting to use these products at 13–19 years, while 57.6% were ≥20 years old. Nearly half (45.4%) of participants did not know/could not remember if they used a no-lye or lye-containing relaxer, 16.0% reported using no-lye relaxers, 21.7% reported using lye-containing relaxers, and 13.3% reported using both. Notably, participants from Embu more frequently reported having their hair relaxed at salons, while those from Nakuru more frequently reported at-home kit use or both. Lastly, almost half of participants reported only using one brand of relaxer products (43.8%) and less than one-third reported using two or more brands (28.4%).

### 3.3. Perceptions about Health Effects and Breast Cancer Risk Related to Hair Product Use

With regards to perceptions about health effects related to hair product use, overall, participants largely agreed (48.1%) or strongly agreed (23.3%) that “organic, natural, non-toxic, or eco-friendly personal hair care products have fewer toxic chemicals than regular products” and largely agreed (48.0%) or strongly agreed (42.0%) that “consumers should be concerned about the health effects of personal hair care products” (Figure 1, Appendix A). Conversely, participants frequently disagreed (39.0%) or strongly disagreed (32.4%) with the notion that “there is no reason to worry about the health effects from chemicals that might be in personal hair care products”. When it came to the statement, “overall, the benefits of using personal hair care products outweigh any risks from exposure to toxic chemicals that might be in these products”, participants in the overall sample were split, with 37.9% agreeing and 23.1% disagreeing. Responses to the BHBS indicated relatively high perceptions of harm/concern around breast cancer risk related to hair product use (Figure 1, Appendix A).

### 3.4. Associations between Participant Characteristics and Hair Dye and Relaxer Use

Increasing age was associated with ever using hair dyes and ever using relaxers in the overall sample (hair dyes: OR = 1.04; 95% CI: 1.02–1.06; relaxers: OR = 1.03; 95% CI: 1.01–1.06) and in Nakuru County (hair dyes: OR = 1.06; 95% CI: 1.02–1.10; relaxers: OR = 1.05; 95% CI: 1.01–1.08) (Table 3). In the overall sample, participants working in sales and service, predominantly cosmetologists, had 105% higher odds of ever using hair dyes (OR = 2.05; 95% CI: 1.38–3.03) and 93% higher odds of ever using relaxers (OR = 1.93; 95% CI: 1.30–2.87) compared to participants working in business and administration. Among participants from Embu, those employed in sales and service had 153% and 105% higher odds of hair dye use (OR = 2.53; 95% CI: 1.47–4.35) and relaxer use (OR = 2.05; 95% CI: 1.19–3.55).

### 3.5. Associations between Perceptions of Harm and Hair Dye Use

Participants in the overall sample who strongly agreed that “organic, natural, non-toxic, or eco-friendly personal hair care products have fewer toxic chemicals than regular products” had 76% higher odds of ever using hair dyes (OR = 1.76; 95% CI: 1.16–2.69) compared to those who responded otherwise (Figure 2). When examining this association stratified by county, the effect became attenuated for participants from Nakuru (OR = 1.25; 95% CI: 0.70–2.23) but doubled in magnitude for those from Embu (OR = 2.60; 95% CI: 1.35–5.00). Furthermore, Nakuru participants who strongly disagreed with the statement “government agencies in Kenya do a good job of regulating personal care products to ensure they are safe for consumers” had 49% lower odds of ever using hair dyes (OR = 0.51; 95% CI: 0.25–1.04) compared to those who expressed otherwise. No comparable association was observed among Embu participants who strongly disagreed with this statement (OR = 1.15; 95% CI: 0.49–2.66).

With regards to perceived breast cancer risk in relation to hair product use, participants in the overall sample who strongly agreed that “[they] want to learn more about the risk hair products can cause to [their] health” had 88% higher odds of ever using hair dyes (OR = 1.88; 95% CI: 1.26–2.82) compared to those who responded otherwise. This positive association persisted upon stratification by county (Embu: OR = 2.13; 95% CI: 1.14–3.97; Nakuru: OR = 1.70; 95% CI: 0.96–3.01). Additionally, participants in the overall sample who strongly agreed that “because [they are] concerned about breast cancer, [they] plan to adjust how [they] use hair care products” had 32% lower odds of hair dye use (OR = 0.68; 95% CI: 0.45–1.04) than those who responded otherwise.

We observed no associations between perceptions about health effects and perceived breast cancer risk related to hair product use with current hair dye use in the overall sample or by county (Appendix A).

### 3.6. Associations between Perceptions of Harm and Relaxer Use

We observed no significant associations between perceptions about health effects and perceived breast cancer risk related to hair product use and ever using relaxers in the overall sample, but county-specific associations were evident (Figure 3). Embu participants who strongly agreed that “consumers should be concerned about the health effects of personal hair care products” had 88% higher odds of relaxer use (OR = 1.88; 95% CI: 1.09–3.22) compared to those who responded otherwise. Also, Embu participants who strongly agreed that “[they] want to learn more about the risk hair products can cause to [their] health” had 73% higher odds of regular relaxer use (OR = 1.73; 95% CI: 0.95–3.18) than those who expressed otherwise. This differed from those in Embu who strongly agreed with the statement “I am concerned that the labels of hair care products do NOT list all the ingredients”, as these individuals had 49% lower odds of regular relaxer use (OR = 0.51; 95% CI: 0.28–0.93) than those that responded otherwise. As for Nakuru participants, those who strongly disagreed with the statement “overall, the benefits of using personal hair care products outweigh any risks from exposure to toxic chemicals that might be in these products” had 125% higher odds of ever using relaxers (OR = 2.25; 95% CI: 1.05–4.82) than those who responded otherwise.

Regarding current relaxer use, positive associations were observed among those who strongly agreed with the statement “consumers should be concerned about the health effects of personal hair care products” in the overall sample (OR = 1.55; 95% CI: 1.06–2.27), and specifically in Embu County (OR = 2.08; 95% CI: 1.23–3.52) (Appendix A). Furthermore, in the overall sample (OR = 2.06; 95% CI: 1.20–3.54) and specifically in Nakuru County (OR = 2.63; 95% CI: 1.25–5.53), participants who strongly disagreed that “government agencies in Kenya do a good job of regulating personal care products to ensure they are safe for consumers” had higher odds of current relaxer use.

With respect to perceived breast cancer risk, participants who strongly agreed that “because [they are] concerned about breast cancer, [they] plan to go natural (style my hair without chemicals)” had 44% and 49% lower odds of current relaxer use, respectively, in the overall sample (OR = 0.56; 95% CI: 0.36–0.86) and in Embu County (OR = 0.51; 95% CI: 0.29–0.91). Meanwhile, those who strongly agreed with the statement “because I am concerned about breast cancer, I plan to adjust how I use hair care products” had 67% higher odds of current relaxer use in the overall sample (OR = 1.67; 95% CI: 1.09–2.57), and 94% higher odds in Nakuru County (OR = 1.94; 95% CI: 1.03–3.66).

## 4. Discussion

Among a large sample of Kenyan women, diverse in age, education, occupation, and income, we found that approximately one-third of participants reported ever using permanent (31.1%) and semi-permanent (37.0%) hair dyes and approximately one-fifth reported current use of permanent hair dyes (17.6%) and semi-permanent (21.3%) hair dyes. Considering combined use of both hair dye types, a little more than half of participants overall (53.5%) were classified as regular users (i.e., inclusive of permanent and/or semi-permanent hair dye use more than twice per year for ≥1 year), with evidence of greater prevalence among participants from the more urban than rural county/areas sampled (55.9% in Nakuru vs. 51.1% in Embu). In terms of relaxer use, more than half the participants were classified as regular users (59.4%) and 35.7% reported current use of these products, with no observable differences in use by county. We also utilized robust measures to evaluate the attitudes and perceptions regarding the link between hair dye and relaxer use and breast cancer risk among this understudied population, creating a unique opportunity to assess exposure risk and factors affecting use. Reported perceptions and attitudes about hair product use and the associated health effects and concern about breast cancer suggest that women included in this study generally have moderately higher levels of concern about the potential health risks associated with using hair products and other PCPs than reported in US-based studies, including our recent analysis of a sample of university-affiliated adults [15,17]. However, we observed mixed results about whether perceptions indicative of higher levels of concern around product use associate with lower odds of hair dye and/or relaxer use.

Hair dye use is ubiquitous among femme-identifying individuals globally and for the first time, we report that this may also be true among Kenyan women, as one-third of our participants reported ever using hair dyes and approximately one-fifth reported currently using these products. While most B/AA women in US-based studies report using hair products at younger ages (e.g., approximately 40% reporting relaxer use before age 12, and approximately 80% reporting relaxer use at age 13–19 years [9]), we found that most women in the current study reported that they started using hair dyes at ages 20 years or older (approximately 70%). Similarly, regular use (approximately 60%) and current use of relaxers (approximately 36%) were quite prevalent in this sample, as reported in another study surveying women in Nakuru [13]. Elsewhere in Africa, few studies have reported on the prevalence of hair dye use [20] and relaxer use [21]. A cross-sectional study in Nigeria—which assessed factors associated with hair greying [20]—reported a prevalence of hair dye use of approximately 16% among individuals with grey hair. Among the sample of control participants in the Ghana Breast Health Study—a breast cancer case–control study—the prevalence of chemical relaxer use for at least one year was 90% [21]. In the Ghanaian study, most participants who used relaxers reported first using them after age 21 years (67%), which is consistent with our findings among Kenyan women (e.g., approximately 70% reporting use at ages 20 years or older) [21]. As earlier exposure to potentially toxic chemicals—including EDCs—found in hair products constitutes an important risk factor for hormone-related conditions [22], our finding that initiation of hair dye and relaxer use occurs later among Kenyan women is important as it may suggest the exposure burden in this population occurs later in the life course, which may be beneficial. A recent review study concluded that exposure to the EDCs commonly found in these products in utero, during puberty, and during pregnancy, may result in alterations in mammary gland development [22]. Exposures during these critical periods of susceptibility may be associated with increased risk of breast cancer, linking the role of earlier hair product use to early onset breast cancer in B/AA women in the US [22].

Significant predictors of both hair dye and relaxer use included increasing age and working in sales and service, especially in the field of cosmetology. Additionally, participants who strongly agreed that “[they] want to learn more about the risk hair products can cause to [their] health” had higher odds of ever using hair dyes. Meanwhile, those who strongly agreed with the statement “because I am concerned about breast cancer, I plan to adjust how I use hair care products” had higher odds of current relaxer use in the overall sample. These findings suggest that targeted interventions and education on hair product use, chemical exposures associated with use and their potential impact on health outcomes, and safe use would largely benefit older women and especially cosmetologists, who may serve as avenues to spread awareness to their clientele. Increasing knowledge of hair product use and safety may benefit Kenyan women with limited knowledge of the chemicals used in their hair and increase environmental health literacy, especially when most participants reported not knowing/remembering the type of chemical relaxer (no-lye vs. lye-containing) they use regularly (46.8% and 43.8% in Embu and Nakuru counties, respectively). One community-based participatory research study concluded that targeted interventions focusing on chemical exposures through cosmetics are an effective measure to increase environmental health literacy as a means to promote healthier PCP use behaviors [23]. Similar interventions emphasizing hair product use in this population may result in more informed health decisions, particularly among participants who expressed interest in learning more about the risk hair products can cause to their health (96.1%). Furthermore, because most participants reported receiving their hair dye or relaxer treatments at salons (59.9% of permanent hair dye users, 56.9% of semi-permanent hair dye users, and 71.6% of relaxer users in the overall sample), increasing awareness (e.g., on the risks of repeated exposure to the chemicals found in these treatments) among cosmetologists may aid in environmental health literacy in this population.

Perceptions of health effects and breast cancer risk among this sample of Kenyan women mirrored those of our US-based study [15] of PCP use among a socioeconomically diverse sample of individuals at a US research university in several ways. In both studies, participants agreed with the statement “consumers should be concerned about the health effects of personal [hair] care products”. Participants also both largely disagreed that “there is no reason to worry about the health effects from chemicals that might be in personal [hair] care products”, with Black/AA women expressing a lower level of agreement in the US-based sample [15]. Kenyan participants also largely agreed or strongly agreed that “organic, natural, non-toxic, or eco-friendly personal [hair] care products have fewer toxic chemicals than regular products”, and participants in our earlier study moderately agreed with this statement as well [15]. We observed few variations in perceptions about product use between participants in Kenya and the US. For the statement “the personal hair care products I use affect my health”, agreement levels varied across county in the Kenyan population. Embu County respondents more frequently agreed (22.8% vs. 15.8%) or strongly agreed (8.1% vs. 2.9%) with this statement, while Nakuru County participants frequently disagreed (48.7% vs. 36.3%). On average, participants in the US-based study moderately agreed that the PCPs they use affect their health [15]. Finally, when asked whether government agencies do a good job of regulating personal care products to ensure they are safe for consumers, responses were mixed. Participants from Embu more frequently strongly agreed (35.2% vs. 20.9%), while those from Nakuru more frequently disagreed (15.5% vs. 9.1%), and US participants trended towards moderate agreement, with females agreeing slightly less than males [15].

Across both populations, Black/AA women expressed concern about the health effects of and chemicals in personal and hair care products. As the most frequent users of hair care products [14,15] and a population at high risk of breast cancer with more aggressive tumor phenotypes [6,7,8,9], education on hair product safety is imperative. While attitudes about whether the hair care products participants use are mixed, further investigation of product use patterns and perceptions of risk must be carried out in other Black/AA subgroups as there is disproportionate exposure to carcinogenic EDCs in these populations.

This study has some limitations that should be considered, including a cross-sectional study design which prevented an analysis of changes in perceptions and product use over time, and the potential for recall bias and/or social desirability bias. Additionally, this study was conducted in only two counties in Kenya; thus, it is possible that participants’ hair dye and relaxer use behaviors and perceptions around use differ from individuals in other regions of the country. Relatedly, only femme-identifying, cisgender women were included in this study, limiting our understanding of PCP and hair product use and exposure risk among sexual and gender minority groups including non-binary and transgender women who are also at risk of breast cancer. Finally, it is of note that study participants were willing to spend 20–30 min to complete a questionnaire asking about their product use and attitudes on health and safety, and these individuals may have prior interest or concern about the impact of using PCPs and hair products on their health.

## 5. Conclusions

Despite these limitations, this study documented for the first time the prevalence and characteristics of hair product use and perceptions around the use of these products among women in Embu and Nakuru Counties, Kenya. Our findings suggest that widespread use of both hair dyes and relaxer use might be an important source of exposure to harmful EDCs commonly linked to increased risk of breast cancer and other hormone-related cancers. Findings were mixed regarding the associations of higher levels of concern around product use with reduced likelihood of hair dye and relaxer use, though many perceptions align with findings from women in the US. Future studies are warranted to investigate the prevalence and patterns of product use and strategies for increasing knowledge and awareness about potential health risks in gender expansive samples and in other countries in Sub-Saharan Africa, where use of these products is widespread and where there is little to no consumer product regulation.

## Figures and Tables

**Figure 1 ijerph-21-00846-f001:**
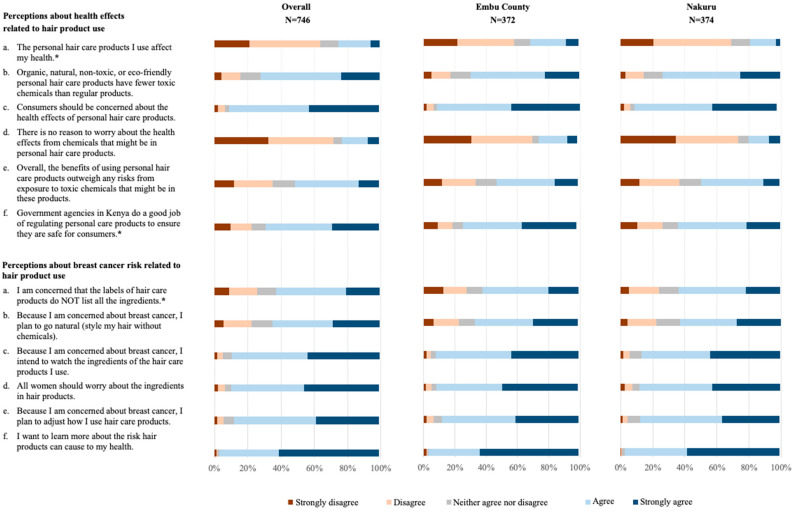
Perceptions about health effects and perceived breast cancer risk related to hair product use among female study participants in Kenya, overall and stratified by county. Stacked bar charts were used to visually depict the percentage of female study participants who strongly agree, agree, neither agree nor disagree, disagree, or strongly disagree with (i) perceptions about health effects related to hair product use (6 items) and (ii) perceived breast cancer risk related to hair product use (6 items), in the overall sample and stratified by county (Embu vs. Nakuru). The two-sided Chi-Squared test was used to assess differences in study participants’ 5-point Likert scale responses across counties. The Benjamini-Hochberg (BH) approach was employed to correct for multiple comparisons, using a False Discovery Rate (FDR) of 0.05. * Differences in participant responses across counties for the following perception items (marked with an asterisk) remained statistically significant after B-H correction: “The personal hair care products I use affect my health” (*p* < 0.001), “Government agencies in Kenya do a good job of regulating personal care products to ensure they are safe for consumers” (*p* < 0.001), and “I am concerned that the labels of hair care products do NOT list all the ingredients” (*p* = 0.008).

**Figure 2 ijerph-21-00846-f002:**
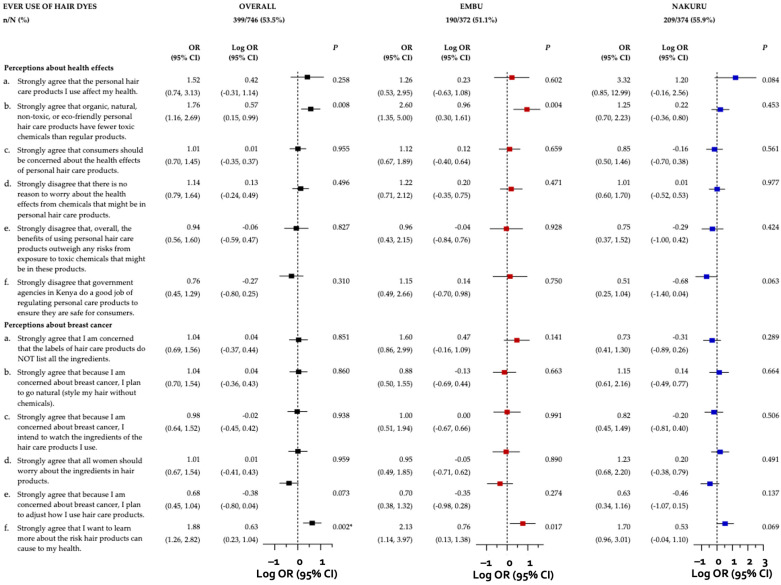
Adjusted odds ratios and 95% confidence intervals illustrating the associations between perceptions about health effects and perceived breast cancer risk and ever use of hair dyes among women in Kenya, overall and stratified by county. Independent multivariable-adjusted logistic regression models were run with robust error variances to examine the association between perceptions about health effects and perceived breast cancer risk related to hair product use (strongly agree/strongly disagree to a perception vs. all other responses) with regular ever hair dye use, overall and stratified by county (Embu vs. Nakuru). Regular ever hair dye use is defined as the use of permanent and/or semi-permanent hair dye(s) more than two times per year for ≥1 year. Each logistic regression model adjusted for relevant sociodemographic (age, education level, current occupation field, number of children under age 15 residing in household) and health history (age at menarche) characteristics. Associations were reported as adjusted odds ratios (ORs) and plotted as adjusted log odds ratios (log ORs), accompanied by their corresponding 95% confidence intervals (CIs). The Benjamini-Hochberg (B-H) approach was employed to correct for multiple comparisons, using a False Discovery Rate (FDR) of 0.05. Following B-H correction, associations with *p*-values ≤0.002 (marked with an asterisk) remained statistically significant.

**Figure 3 ijerph-21-00846-f003:**
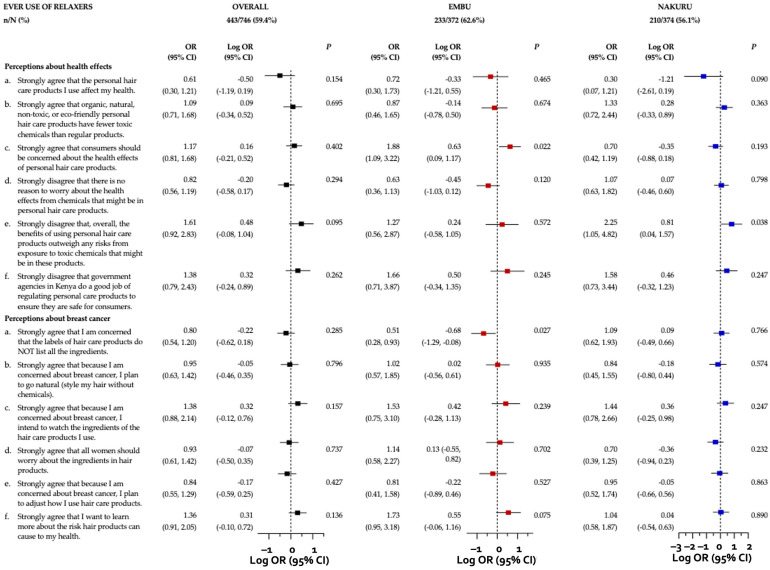
Adjusted odds ratios and 95% confidence intervals illustrating the associations between perceptions about health effects and perceived breast cancer risk and ever use of relaxers among women in Kenya, overall and stratified by county. Independent multivariable-adjusted logistic regression models were run with robust error variances to examine the association between perceptions about health effects and perceived breast cancer risk related to hair product use (strongly agree/strongly disagree to a perception vs. all other responses) with regular ever relaxer use, overall and stratified by county (Embu vs. Nakuru). Regular ever relaxer use is defined as the use of chemical relaxer(s) or straightening product(s) more than two times per year for ≥1 year. Each logistic regression model adjusted for relevant sociodemographic (age, education level, current occupation field, number of children under age 15 residing in household) and health history (age at menarche) characteristics. Associations were reported as adjusted odds ratios (ORs) and plotted as adjusted log odds ratios (log ORs), accompanied by their corresponding 95% confidence intervals (CIs). The Benjamini-Hochberg (B-H) approach was employed to correct for multiple comparisons, using a False Discovery Rate (FDR) of 0.05. Following B-H correction, associations with *p*-values ≤0.002 would be statistically significant, but none were.

**Table 1 ijerph-21-00846-t001:** Characteristics of study participants in Embu and Nakuru Counties, Kenya, overall and stratified by county.

Characteristics	Overall N = 746 N (%)	Embu County N = 372 N (%)	Nakuru County N = 374 N (%)	Embu County vs. Nakuru County *p* ^†^	MISSING ^‡^ N (%)
Sociodemographics					
Age (years), mean ± SD	30.4 ± 8.1	31.1 ± 8.4	29.8 ± 7.8	0.027	1 (0.1)
Religion				0.608	0 (0.0)
Christian	739 (99.1)	369 (99.2)	370 (98.9)		
Muslim	6 (0.8)	3 (0.8)	3 (0.8)		
Other	1 (0.1)	0 (0.0)	1 (0.3)		
Marital status				0.029	2 (0.3)
Currently married	348 (46.6)	178 (47.8)	170 (45.5)		
Formerly married ^a^	52 (7.0)	34 (9.1)	18 (4.8)		
Never married	344 (46.1)	159 (42.7)	185 (49.5)		
Education level				0.224	1 (0.1)
Less than high school certificate	142 (19.0)	81 (21.8)	61 (16.3)		
High school certificate	283 (37.9)	141 (37.9)	142 (38.0)		
Some college but no degree	253 (33.9)	117 (31.5)	136 (36.4)		
Bachelor’s degree and above ^b^	67 (9.0)	32 (8.6)	35 (9.4)		
Current occupation field				<0.001 *	6 (0.8)
Business and administration	155 (20.8)	97 (26.1)	58 (15.5)		
Education, government, and law enforcement	34 (4.6)	14 (3.8)	20 (5.3)		
Farming and agriculture	60 (8.0)	36 (9.7)	24 (6.4)		
Sales and service ^c^	364 (48.8)	155 (41.7)	209 (55.9)		
Science, technology, and engineering	17 (2.3)	8 (2.2)	9 (2.4)		
Student	61 (8.2)	24 (6.5)	37 (9.9)		
Unemployed or casual work	49 (6.6)	35 (9.4)	14 (3.7)		
Current income (KES per month) ^d^				0.003 *	28 (3.8)
<10,000	427 (57.2)	194 (52.2)	233 (62.3)		
10,000–50,000	279 (37.4)	153 (41.1)	126 (33.7)		
>50,000	12 (1.6)	10 (2.7)	2 (0.5)		
Household size				0.074	6 (0.8)
1–2	159 (21.3)	76 (20.4)	83 (22.2)		
3–4	358 (48.0)	194 (52.2)	164 (43.9)		
≥5	223 (29.9)	100 (26.9)	123 (32.9)		
Number of children <15 years residing in household				0.002 *	9 (1.2)
0	186 (24.9)	103 (27.7)	83 (22.2)		
1	242 (32.4)	133 (35.8)	109 (29.1)		
2	210 (28.2)	99 (26.6)	111 (29.7)		
≥3	99 (13.3)	34 (9.1)	65 (17.4)		
Health history					
Age at menarche (years), mean ± SD	14.3 ± 1.7	14.2 ± 1.8	14.5 ± 1.7	0.047	12 (1.6)
Menopausal status ^e^				0.142	5 (0.7)
Premenopausal	654 (87.7)	333 (89.5)	321 (85.8)		
Postmenopausal	87 (11.7)	37 (9.9)	50 (13.4)		
Personal history of breast cancer				0.018	2 (0.3)
No	726 (97.3)	367 (98.7)	359 (96.0)		
Yes	18 (2.4)	4 (1.1)	14 (3.7)		
Currently pregnant				0.630	11 (1.5)
No	688 (92.2)	347 (93.3)	341 (91.2)		
Yes	47 (6.3)	22 (5.9)	25 (6.7)		

Abbreviations: SD, standard deviation. ^†^ For all categorical variables, comparisons of proportions across counties were assessed using the two-sided chi-squared test. For continuous variables, comparisons of means across counties were assessed using analysis of variance (ANOVA). The Benjamini–Hochberg (B–H) approach was employed to correct for multiple comparisons, using a false discovery rate (FDR) of 0.05. Following B–H correction, comparisons with *p*-values ≤ 0.008 (marked with an asterisk) remained statistically significant. ^‡^ The count and proportion missing pertain to the overall sample. ^a^ Formerly married individuals (overall: N = 52, 7.0%) include those who are divorced (overall: N = 7, 0.9%), separated (overall: N = 32, 4.3%), and widowed (overall: N = 13, 1.7%). ^b^ Individuals who completed a bachelor’s degree and above (overall: N = 67, 9.0%) primarily consisted of those who completed a bachelor’s degree (overall: N = 65, 8.7%), with the exception of two (0.3%) individuals who completed a master’s degree in Embu County. ^c^ Individuals in sales and service (overall: N = 364, 48.8%) largely comprised of cosmetologists (overall: N = 302, 40.5%). ^d^ Individuals with missing information on their current income (overall: N = 28, 3.8%) mainly included students or those who were unemployed/had casual work (overall: N = 17, 2.3%). ^e^ Women experiencing menopause reported not having at least one menstrual period in the past 12 months.

**Table 2 ijerph-21-00846-t002:** Characteristics of hair dye and relaxer use among study participants in Embu and Nakuru Counties, Kenya, overall and stratified by county.

	Overall N = 746 N (%)	Embu County N = 372 N (%)	Nakuru County N = 374 N (%)	Embu County vs. Nakuru County *p* ^†^	Missing ^‡^ N (%)
Permanent hair dye use					
Ever using permanent hair dyes regularly for ≥1 year ^a^				0.305	24 (3.2)
No	490 (65.7)	246 (66.1)	244 (65.2)		
Yes	232 (31.1)	107 (28.8)	125 (33.4)		
Age at which regular permanent hair dye use began				0.649	22 (9.5)
≤12 years	2 (0.9)	1 (0.9)	1 (0.8)		
13–19 years	45 (19.4)	17 (15.9)	28 (22.4)		
≥20 years	163 (70.3)	74 (69.2)	89 (71.2)		
Shade of permanent hair dye color predominantly used				<0.001 *	21 (9.1)
Light (blonde, light brown)	50 (21.6)	29 (27.1)	21 (16.8)		
Medium (medium brown, red)	51 (22.0)	35 (32.7)	16 (12.8)		
Dark (dark brown, black)	102 (44.0)	29 (27.1)	73 (58.4)		
Two of the above	8 (3.4)	0 (0.0)	8 (6.4)		
Typical application of permanent hair dye				<0.001 *	22 (9.5)
At-home kit	32 (13.8)	13 (12.1)	19 (15.2)		
Salon	139 (59.9)	73 (68.2)	66 (52.8)		
Both	39 (16.8)	6 (5.6)	33 (26.4)		
Number of different permanent hair dye brands used				0.288	99 (42.7)
1	97 (41.8)	45 (42.1)	52 (41.6)		
≥2	36 (15.5)	13 (12.1)	23 (18.4)		
Current use of permanent hair dye in the past year				0.007 *	71 (9.5)
No	544 (72.9)	270 (72.6)	274 (73.3)		
Yes	131 (17.6)	48 (12.9)	83 (22.2)		
Semi-permanent hair dye use					
Ever using semi-permanent hair dyes regularly for ≥1 year ^a^				0.600	23 (3.1)
No	447 (59.9)	226 (60.8)	221 (59.1)		
Yes	276 (37.0)	134 (36.0)	142 (38.0)		
Age at which regular semi-permanent hair dye use began				0.292	31 (11.2)
≤12 years	1 (0.4)	1 (0.7)	0 (0.0)		
13–19 years	51 (18.5)	20 (14.9)	31 (21.8)		
≥20 years	193 (69.9)	93 (69.4)	100 (70.4)		
Shade of semi-permanent hair dye color predominantly used				0.007 *	32 (11.6)
Light (blonde, light brown)	40 (14.5)	22 (16.4)	18 (12.7)		
Medium (medium brown, red)	46 (16.7)	27 (20.1)	19 (13.4)		
Dark (dark brown, black)	149 (54.0)	65 (48.5)	84 (59.2)		
Two of the above	9 (3.3)	0 (0.0)	9 (6.3)		
Primary location of semi-permanent hair dye use				0.037	34 (12.3)
At-home kit	36 (13.0)	17 (12.7)	19 (13.4)		
Salon	157 (56.9)	81 (60.4)	76 (53.5)		
Both	49 (17.8)	15 (11.2)	34 (23.9)		
Number of different semi-permanent hair dye brands used				0.888	112 (40.6)
1	126 (45.7)	58 (43.3)	68 (47.9)		
≥2	38 (13.8)	17 (12.7)	21 (14.8)		
Current use of semi-permanent hair dye in the past year				0.373	49 (6.6)
No	538 (72.1)	272 (73.1)	266 (71.1)		
Yes	159 (21.3)	74 (19.9)	85 (22.7)		
Chemical relaxer use					
Ever using relaxers regularly for ≥1 year ^a^				0.068	10 (1.3)
No	293 (39.3)	134 (36.0)	159 (42.5)		
Yes	443 (59.4)	233 (62.6)	210 (56.1)		
Age at which regular relaxer use began				0.180	18 (4.1)
≤12 years	20 (4.5)	7 (3.0)	13 (6.2)		
13–19 years	150 (33.9)	85 (36.5)	65 (31.0)		
≥20 years	255 (57.6)	134 (57.5)	121 (57.6)		
Use of no-lye or lye-containing relaxer				0.489	16 (3.6)
No-lye	71 (16.0)	33 (14.2)	38 (18.1)		
Lye-containing	96 (21.7)	49 (21.0)	47 (22.4)		
Both	59 (13.3)	35 (15.0)	24 (11.4)		
I do not know/I cannot t remember	201 (45.4)	109 (46.8)	92 (43.8)		
Primary location of relaxer use				<0.001 *	16 (3.6)
At-home kit	54 (12.2)	23 (9.9)	31 (14.8)		
Salon	317 (71.6)	189 (81.1)	128 (61.0)		
Both	56 (12.6)	14 (6.0)	42 (20.0)		
Number of different relaxer brands used				0.857	123 (27.8)
1	194 (43.8)	95 (40.8)	99 (47.1)		
≥2	126 (28.4)	63 (27.0)	63 (30.0)		
Current use of relaxer in the past year				0.134	31 (4.2)
No	449 (60.2)	212 (57.0)	237 (63.4)		
Yes	266 (35.7)	141 (37.9)	125 (33.4)		

^†^ For all categorical variables, comparisons of proportions across counties were assessed using the two-sided chi-squared test. The Benjamini–Hochberg (B–H) approach was employed to correct for multiple comparisons, using a false discovery rate (FDR) of 0.05. Following B–H correction, comparisons with *p*-values ≤ 0.008 (marked with an asterisk) remained statistically significant. ^‡^ The count and proportion missing pertain to the overall sample. ^a^ By regular use, we mean using the product more than two times per year.

**Table 3 ijerph-21-00846-t003:** Associations between sociodemographic and health history characteristics with ever using hair dyes (inclusive of permanent and/or semi-permanent) and relaxers in Embu and Nakuru Counties, Kenya, overall and by county.

	Overall				Embu County				Nakuru County			
n/N (%)	Hair Dyes 399/746 (53.5%)		Relaxers 443/746 (59.4%)		Hair Dyes 190/372 (51.1%)		Relaxers 233/372 (62.6%)		Hair Dyes 209/374 (55.9%)		Relaxers 210/374 (56.1%)	
	OR (95% CI)	*p* ^†^	OR (95% CI)	*p* ^†^	OR (95% CI)	*p* ^†^	OR (95% CI)	*p* ^†^	OR (95% CI)	*p* ^†^	OR (95% CI)	*p* ^†^
Age (years)	1.04 (1.02, 1.06)	<0.001 *	1.03 (1.01, 1.06)	0.004	1.02 (1.00, 1.05)	0.093	1.01 (0.99, 1.05)	0.331	1.06 (1.02, 1.10)	0.001 *	1.05 (1.01, 1.08)	0.004
Education level												
Less than high school certificate	ref.		ref.		ref.		ref.		ref.		ref.	
High school certificate	1.16 (0.75, 1.80)	0.494	0.91 (0.58, 1.41)	0.665	1.19 (0.66, 2.16)	0.561	0.80 (0.44, 1.46)	0.469	1.09 (0.56, 2.14)	0.798	1.14 (0.59, 2.18)	0.698
Some college but no degree	1.06 (0.67, 1.68)	0.807	1.34 (0.83, 2.16)	0.226	0.98 (0.51, 1.88)	0.953	0.81 (0.41, 1.58)	0.528	0.99 (0.49, 2.00)	0.979	2.44 (1.21, 4.90)	0.012
Bachelor’s degree and above ^a^	0.91 (0.48, 1.73)	0.771	0.85 (0.44, 1.64)	0.626	1.60 (0.63, 4.05)	0.321	0.63 (0.24, 1.63)	0.337	0.46 (0.18, 1.19)	0.110	1.33 (0.50, 3.54)	0.565
Current occupation field												
Business and administration	ref.		ref.		ref.		ref.		ref.		ref.	
Farming and agriculture	0.90 (0.47, 1.70)	0.736	1.13 (0.60, 2.12)	0.709	1.21 (0.52, 2.80)	0.653	1.22 (0.53, 2.79)	0.637	0.63 (0.23, 1.74)	0.370	1.13 (0.41, 3.12)	0.816
Sales and service ^b^	2.05 (1.38, 3.03)	<0.001 *	1.93 (1.30, 2.87)	0.001 *	2.53 (1.47, 4.35)	0.001 *	2.05 (1.19, 3.55)	0.010	1.51 (0.84, 2.72)	0.165	2.21 (1.18, 4.13)	0.013
Student, unemployed, or casual work	1.54 (0.88, 2.67)	0.128	1.59 (0.92, 2.76)	0.096	1.29 (0.63, 2.65)	0.487	1.57 (0.75, 3.30)	0.232	1.89 (0.77, 4.60)	0.164	1.67 (0.72, 3.86)	0.232
Other ^c^	1.51 (0.76, 3.02)	0.241	1.58 (0.76, 3.27)	0.221	1.81 (0.64, 5.13)	0.266	3.00 (0.95, 9.47)	0.061	1.06 (0.40, 2.83)	0.900	1.14 (0.40, 3.24)	0.809
Number of children <15 years residing in household												
0	ref.		ref.		ref.		ref.		ref.		ref.	
1	1.19 (0.80, 1.77)	0.395	1.05 (0.70, 1.58)	0.812	1.35 (0.78, 2.33)	0.280	1.11 (0.64, 1.93)	0.700	0.96 (0.51, 1.79)	0.887	0.99 (0.52, 1.88)	0.973
2	1.05 (0.69, 1.60)	0.826	0.99 (0.65, 1.53)	0.971	1.21 (0.67, 2.19)	0.532	1.09 (0.59, 2.01)	0.789	0.77 (0.41, 1.43)	0.405	0.95 (0.50, 1.83)	0.885
≥3	0.92 (0.55, 1.53)	0.735	1.19 (0.70, 2.04)	0.517	1.38 (0.61, 3.10)	0.440	1.79 (0.74, 4.34)	0.195	0.57 (0.27, 1.17)	0.126	1.19 (0.57, 2.51)	0.638
Age at menarche (years)	0.95 (0.87, 1.04)	0.274	1.02 (0.93, 1.11)	0.670	1.01 (0.90, 1.14)	0.851	1.01 (0.90, 1.14)	0.829	0.87 (0.77, 1.00)	0.044	1.01 (0.88, 1.16)	0.901

^†^ Independent logistic regression models were run with robust error variances to obtain odds ratios (ORs) with corresponding 95% confidence intervals (CIs) that examined the association between sociodemographic and health history characteristics of study participants with regular hair dye use and regular relaxer use, separately, in the overall sample and stratified by county (Embu vs. Nakuru). Regular hair dye use is defined as the use of permanent and/or semi-permanent hair dye(s) more than two times per year for ≥1 year. Regular relaxer use is defined as the use of chemical relaxer(s) or straightening product(s) more than two times per year for ≥1 year. The Benjamini–Hochberg (B–H) approach was employed to correct for multiple comparisons, using a false discovery rate (FDR) of 0.05. Following B–H correction, associations with *p*-values ≤0.002 (marked with an asterisk) remained statistically significant. ^a^ Individuals who completed a bachelor’s degree and above (overall: N = 67, 9.0%) primarily comprised of those who completed a bachelor’s degree (overall: N = 65, 8.7%), with the exception of two (0.3%) individuals who completed a master’s degree in Embu County. ^b^ Individuals in sales and service (N) largely comprised of cosmetologists (n)—overall n/N: 302/364 (83.0%); Embu n/N: 112/155 (72.3%); Nakuru n/N: 190/209 (90.9%). ^c^ The other category combines individuals who belong to the following occupation fields: education, government, law enforcement, science, technology, and engineering.

## Data Availability

The data presented in this study are available upon request from the corresponding authors. The data are not publicly available due to privacy restrictions.

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
