# Peer review of "Hair Dye and Relaxer Use among Cisgender Women in Embu and Nakuru Counties, Kenya: Associations with Perceived Risk of Breast Cancer and Other Health Effects"

_ijerph, 2024, doi:10.3390/ijerph21070846_

Round 1

Reviewer 1 Report

Comments and Suggestions for Authors

Congratulation to interesting and relevant work! I have only the following three issues to ask for and potentially modify:

1. Could you please desribe more in details the sample selection? It is very superficially described....Convenience sampling is fine, convenient at what sense? Source population? What about those women who do not go to cosmetologists? Were they fully excluded?

2. why it is important to stress that the sample is cis-gender women? Is there any difference in biological pathway of EDC's on hormonal mechanisms by different gender groups of women? Seems to be irrelevant to me...

3. Could you please simplify the results? there is a lot of information in tables and text and it becomes hard to read....

Reviewer 2 Report

Comments and Suggestions for Authors

Round 2

Reviewer 2 Report

Comments and Suggestions for Authors

No further comments.